# Metal Ions Modify In Vitro DNA Damage Yields with High-LET Radiation

**DOI:** 10.3390/toxics11090773

**Published:** 2023-09-12

**Authors:** Dylan J. Buglewicz, Cathy Su, Austin B. Banks, Jazmine Stenger-Smith, Suad Elmegerhi, Hirokazu Hirakawa, Akira Fujimori, Takamitsu A. Kato

**Affiliations:** 1National Institute of Radiological Sciences, National Institutes of Quantum Science and Technology, Chiba 263-8555, Japan; dbuglewi@rams.colostate.edu (D.J.B.); hirakawa.hirokazu@qst.go.jp (H.H.); fujimori.akira@qst.go.jp (A.F.); 2Department of Environmental & Radiological Health Sciences, Colorado State University, Fort Collins, CO 80523, USA; cathy50720@gmail.com (C.S.); abbanks@rams.colostate.edu (A.B.B.); jazmine.stengersmith@gmail.com (J.S.-S.); hanoyara@rams.colostate.edu (S.E.)

**Keywords:** high-LET radiation, DNA breaks, Fenton reaction, carbon-ion radiation, DNA damage

## Abstract

Cu^2+^ and Co^2+^ are metals known to increase DNA damage in the presence of hydrogen peroxide through a Fenton-type reaction. We hypothesized that these metals could increase DNA damage following irradiations of increasing LET values as hydrogen peroxide is a product of the radiolysis of water. The reaction mixtures contain either double- or single-stranded DNA in solution with Cu^2+^ or Co^2+^ and were irradiated either with X-ray, carbon-ion or iron-ion beams, or they were treated with hydrogen peroxide or bleomycin at increasing radiation dosages or chemical concentrations. DNA damage was then assessed via gel electrophoresis followed with a band intensity analysis. DNA damage was the greatest when DNA in the solution with either metal was treated with only hydrogen peroxide followed by the DNA damage of DNA in the solution with either metal post irradiation of low-LET (X-Ray) or high-LET (carbon-ion and iron-ion), respectively, and demonstrated the least damage after treatment with bleomycin. Cu^2+^ portrayed greater DNA damage than Co^2+^ following all experimental conditions. The metals’ effect caused more DNA damage and was observed to be LET-dependent for single-strand break formation but inversely dependent for double-strand break formation. These results suggest that Cu^2+^ is more efficient than Co^2+^ at inducing both DNA single-strand and double-strand breaks following all irradiations and chemical treatments.

## 1. Introduction

The major actor responsible for radiation-induced cell death is DNA double-strand breaks (DSBs) from localized single-strand breaks (SSBs). Generally, 1 Gy of radiation produces 20–40 DSBs and a few thousands of SSB in cells [1]. Radiation-induced DNA damage is caused by direct action or indirect action through the radiolysis of water. X-rays are known as low-LET (linear energy transfer) radiation and are sparsely ionizing; [2] a large amount of DNA damage is believed to be contributed by hydroxyl radical mediated indirect action. However, radiation-induced cell death is not due to individual hydroxyl radicals but the interaction of radicals at high density near DNA causing locally multiply damaged sites (LMDS) [3]. Indeed, high-LET radiation such as neutrons and heavy charged particles, are known to be densely ionizing and produce higher biological effectiveness [2]. This is because there is an increase in direct action contribution for high-LET induced DNA damage through stronger electromagnetic interactions with the DNA molecule [4,5]. The dense ionization produced by high-LET radiation creates high radical concentrations from the radiolysis of water leading to more radical–radical reactions and resulting in the formation of greater G-values of hydroxyl radicals [6].

Yields of radicals in solution can be affected by factors including pH, temperature, solvents and substrates [7,8]. As the hydroxyl radical scavengers, such as dimethyl sulfoxide (DMSO), attenuate radiation effects [9], the amount of hydroxyl radicals can interfere with DNA damage yields. This is because metal ions and hydrogen peroxide can cause Fenton reactions, thus producing greater amounts of hydroxyl radicals [10]. Therefore, the interaction among radicals can be interfered with by adding metal ions and may maximize the indirect action of radiation to enhance the radiation effects on DNA damage. Superoxide dismutase and Catalase are endogenous enzymes that reduce final hydroxyl radical formation [11]. Interestingly, another DNA DSB-inducing agent, bleomycin, has a metal-binding domain and requires metals and oxygen to cleave DNA [12,13,14,15]. Bleomycin and the metal complex create radicals and contribute to DNA damage. Various studies support adding different metals to increase radiation or drug induced hydroxyl radical formation and DNA breaks [7,16,17,18,19].

In order to better understand the underlying mechanisms behind DNA damage induced by hydroxyl radicals in high-LET radiation and investigate potential sensitization through enhancement of indirect action, we took advantage of Fenton-type reactions utilizing the metals copper and cobalt. Prior studies have observed an increase in DNA strand breaks and a decrease in the molecular weight of DNA in mammalian cells treated with either copper or cobalt [20], and these metals in complexes have also been observed to decrease survival following irradiation [16,17]. Moreover, Lloyd D.R. et al. suggested that the site-specific mechanisms in the formation of DSBs were likely due to the oxidative DNA damaged mediated by copper Fenton reactions in solution with DNA and hydrogen peroxide. Prior work from Lloyd D.R. et al. also demonstrated ^32^P-Postlabelling analysis of DNA treated with hydrogen peroxide and either copper(II), cobalt (II), chromium (VI), iron (II), nickel (II), or vanadium (III), which resulted in the detection of between four and eight radioactive TLC spots corresponding to DNA lesions and, importantly, the copper-Fenton system generated the highest total yield of these DNA lesions followed by cobalt-Fenton system [19]. Thus, we hypothesized that the copper ions and cobalt ions would be useful chemicals to increase DNA strand breaks. To test our hypothesis, we used in vitro single- and double-stranded DNA and three different qualities of radiation with different LET values and two major chemicals known to cause DNA damage to assess each metal ion’s effect for inducing DNA damage.

## 2. Materials and Methods

### 2.1. Irradiation Conditions

Low-LET irradiations were conducted utilizing the X-ray generator TITAN 320 (200 kVp, 20 mA, 5 mm aluminum and copper filter). X-ray exposure rate was 3.1 Gy/min, and LET value of 2 keV/μm [21]. For high-LET heavy ion irradiations, spread out Bragg-peak (SOBP) carbon-ions and monoenergetic iron-ions were accelerated to 290 and 500 MeV/n, respectively, using HIMAC. Dose rates for carbon-ions and iron-ions were set at 5 and 10 Gy/min, respectively. SOBP carbon-ions and monoenergetic iron-ions contained a LET value of 50 and 200 keV/μm, respectively [21].

### 2.2. DNA Solution Preparation and Chemical Treatment

A total of 10 μL of reaction solution was used containing 30 ng of double-strand DNA (dsDNA) of lambda phage (New England BioLabs Inc., Ipswich, MA, USA, stock concentration of 500 ng/μL, 48,502 base pairs in length, N3011S) or 83 ng of single-strand DNA (ssDNA) of M13mp18 phage (New England BioLabs Inc, stock concentration of 250 ng/μL, 7249 bases in chain length, N4040S) with 10 mM Tris-HCl pH 7.71 with or without 0.2 mM of CuCl_2_ or CoCl_2_.

For chemical treatment experiments, hydrogen peroxide (Sigma-Aldrich, Merk kGaA, Darmstadt, Germany) or bleomycin (Sigma-Aldrich) was added to a total of 10 μL of reaction solution with or without metals. Once the solution was made, they were incubated at 37 °C for 30 min. Following irradiation or chemical treatment incubation, 1 mM of EDTA was added to chelate excess metals within the solution and incubated at room temperature for 5 min [7].

### 2.3. Electrophoresis and DNA Damage Quantification

Agarose gel electrophoresis was carried out as previously described [22,23,24]. Each sample was added with 6× loading dye (15% Ficoll (*w*/*v*), 10% glycerol (*v*/*v*), 0.25% bromophenol blue (*w*/*v*) and 0.25% xylene cyanol FF (*w*/*v*) in distilled water) and electrophoresis was carried out with an 1% (*w*/*v*) Agarose gel in 1X TAE buffer containing 0.01% (*w*/*v*) ethidium bromide and ran at 100 V and for 60 min in 1x TAE buffer. After electrophoresis and destaining in distilled water, gels were imaged using Bio-Rad Universal Hood II (Bio-Rad Laboratories, Hercules, CA, USA), and band intensity measurements were processed via Image Lab^TM^ software version 2.0.1 (Bio-Rad Laboratories). Band intensities of intact DNA were then normalized to control that were not irradiated nor treated with chemicals and calculated fraction of intact DNA after irradiation or chemical treatment.

D_50_ and IC_50_ values, dose or chemical concentration required to produce 50% intact DNA, were determined using regression curves generated using GraphPad Prism 8 (GraphPad Software, San Diego, CA, USA). From these values, metal enhancement ratios (MER) were calculated from D_50_ (IC_50_) of control values divided by D_50_ (IC_50_) of tested agent values.

### 2.4. Statistical Analysis

All experimental data were derived from at least three independent experiments. Data points were expressed as a mean with standard errors of the means. Statistical significance was determined using one-way analysis of variance (ANOVA) followed by Tukey’s multiple comparison test using GraphPad Prism 8^TM^ software (GraphPad, La Jolla, CA, USA). *p* < 0.05 was considered statistically significant.

## 3. Results

### 3.1. Copper and Cobalt Ions Increased DNA DSB Following Ionizing Radiation

DSB formation yield measured with 30 ng of in vitro dsDNA from lambda phage in 10 µL of reaction solution was higher in low-LET X-ray and decreased as LET value increased in carbon-ion and iron-ion, respectively. D_50_ values (dose to achieve 50% intact DNA) were achieved with 44, 142 and 299 Gy, respectively. X-ray irradiation produced a significant increase in DSB with both metals compared to control. Carbon-ion irradiation produced a significant increase in DSB with adding Cu^2+^ or Co^2+^, compared to control. Adding Cu^2+^ produced a significant increase in DSB formation compared to adding Co^2+^ (Figure 1b and Appendix A). Finally, iron-ion irradiation demonstrated an increase in DSB with the addition of both metals, most notably with Cu^2+^, but this observed difference was not found to be significant under all experimental dosages (Figure 1c).

**D_50, DSB,_** demonstrated fold increases of efficiency of DSB with addition of metal ions (Table 1). **D_50, DSB_** values with addition of Cu^2+^ were smaller than adding Co^2+^ for all tested radiations. Therefore, adding Cu^2+^ was more efficient at increasing DSB yields for tested radiation than adding Co^2+^.

### 3.2. Copper and Cobalt Ions Increased DNA SSB Following Ionizing Radiation

Induction of SSB was measured with 83 ng of in vitro ssDNA originated from M13 bacteriophage in 10 µL of reaction solution. As DSB, X-ray was the most efficient to cause SSB as **D_50, SSB_** values of 57 Gy among tested radiation. **D_50, SSB_** values of high-LET radiation carbon-ion and iron-ion were 130 Gy and 160 Gy, respectively. Irradiation of the X-ray was observed to produce a significant increase in SSB with the addition of Cu^2+^ and Co^2+^. Adding Cu^2+^ and Co^2+^ showed significant increase in SSB formation compared to control (Figure 1d and Appendix A). Carbon-ion irradiation produced a significant increase in SSB formation with adding both metals compared to control (Figure 1e). Then, iron-ion irradiation also produced a significant increase in SSB formation with both metals (Figure 1f).

**D_50, SSB_** values demonstrated fold increases of efficiency of SSB with addition on metal ions (Table 1). **D_50, SSB_** values with adding Cu^2+^ were smaller than adding Co^2+^ for all tested radiation. Therefore, adding Cu^2+^ was more efficient at increasing DSB yields for tested radiation than adding Co^2+^.

### 3.3. Copper and Cobalt Ions Increased DNA DSB When in Solution with Bleomycin or Hydrogen Peroxide

To induce DNA damage, bleomycin or hydrogen peroxide (H_2_O_2_) were treated with DNA at room temperature for 30 min. Bleomycin is known to produce DSB and SSB [12]. Hydrogen peroxide produces SSBs [10]. Bleomycin presented concentration-dependent DSB formation up to 1413 µM with a **D_50, DSB_** value of 46,625 nM (Figure 2a and Appendix A). At 14.1 µM, adding both metals enhanced DSB formation. Interestingly, from 70 μM of bleomycin, adding Co^2+^ was observed to switch from sensitization to protection in comparison to control.

Hydrogen peroxide, up to 10 mM, did not produce a significant amount of DSB without metals. Based on regression curves, IC_50, DSB_ was calculated as 20 mM. By adding either metal ions, hydrogen peroxide treatment was observed to produce a significant increase in DSB formation (Figure 2b).

**IC_50, DSB_** values demonstrated fold increases of efficiency of DNA breaks with addition of metal ions (Table 1)**.** Although **IC_50, DSB_** values of bleomycin were decreased slightly with metals, **D_50, DSB_** values of hydrogen peroxide were severely decreased from 20,073 nM to 22 nM (Cu^2+^) and 177 nM (Co^2+^). Therefore, bleomycin had minimal effects from metal ions to form DSB. On the other hand, the addition of metal ions to solutions with hydrogen peroxide severely affected the amount of DSBs, and copper ion’s effect was observed to be much stronger than with cobalt ions.

### 3.4. Copper and Cobalt Ions Increased DNA SSB When in Solution with Bleomycin or Hydrogen Peroxide

Bleomycin produced SSB in a concentration-dependent manner. As with DSB formation, bleomycin demonstrated a shift from sensitivity to protection with adding Co^2+^ in SSB formation above 7.1 µM. Adding Cu^2+^ was observed to increase the sensitivity of bleomycin compared to control, but these decreases in SSB formation were not observed to be significant (Figure 2c and Appendix A).

Hydrogen peroxide produced dose-dependent SSB formation up to a tested amount of 10 μM. A significant increase in SSB was observed with adding Cu^2+^ at the lowest tested concentration of 100 pM. Adding cobalt ions increased SSBs, but the effect was much smaller than adding copper (Figure 2d).

**IC_50, SSB_** values demonstrated fold increases of efficiency of DNA breaks with addition of metal ions (Table 1). Although **IC_50, SSB_** values of bleomycin showed a few changes by adding metal ions, **D_50, SSB_** values of hydrogen peroxide were dramatically decreased with addition of metals. Therefore, bleomycin had minimal effects from the addition of metal ions to form SSB. On the other hand, as with DSBs, the addition of metal ions to solutions with hydrogen peroxide severely affected the amount of SSBs, and copper ion’s effect was again observed to be much stronger than with cobalt ions.

### 3.5. Metal Enhancement Ratio for DNA Break Formation Was Highest for Hydrogen Peroxide Followed by Ionizing Radiation and Least for Bleomycin

Metal enhancement ratio’s (MER) were calculated from the **D_50_** and **IC_50_** values from Table 1 and described in the Materials and Methods section. For ionizing radiation with dsDNA, the MER demonstrated a fold reduction of 5.18×, 5.09× and 3.74× with Cu^2+^ and 2.84×, 1.77× and 1.6× with Co^2+^ following X-ray, carbon-ion or iron-ion, respectively. Bleomycin showed the smallest MER values. The MER demonstrated a fold reduction of 914.94× with Cu^2+^ and 113.41x with Co^2+^ following H_2_O_2_ (Figure 3a). For ionizing radiation with single-strand DNA, the MER demonstrated a fold reduction of 18.37×, 21.96× and 22.92× with Cu^2+^ and 5.97×, 2× and 1.57× with Co^2+^ following X-ray, carbon-ion and iron-ion, respectively. The MER values for SSB were greater values than for DSB. Bleomycin showed the smallest MER values. The MER demonstrated a fold reduction of 110,493× with Cu^2+^ and 15.15× with Co^2+^ following H_2_O_2_. (Figure 3b). Therefore, metal ions affected both SSB and DSB formation induced by tested chemicals and radiation. Hydrogen peroxide affected DNA damage the most followed by low-LET radiation and high-LET radiation, and bleomycin affected the least. Copper ions presented stronger effects than cobalt ions. SSB formation was enhanced more with metal ions than DSBs. Metal effects for DNA damage were LET-dependent for SSB formation but inversely dependent for DSB formation.

## 4. Discussion

In the present study, we observed that the induction of DSBs and SSBs reduced with increasing LET values as low-LET X-ray irradiations were the most efficient inducers of DNA strand breaks per dose (Gy) followed by the high-LET carbon-ion and iron-ion beams, respectively (Table 1). These results are in agreement with prior research with cell lines, which indicate that the number of SSBs produced is estimated to be lower in high-LET radiation [2]. In contrast to our results, other prior studies have demonstrated an increase in DSBs with increased LET in cells [25,26]. This is because our method of measuring DNA damage utilizing naked DNA in solution with or without metal ions via agarose gel electrophoresis provided several advantages and disadvantages over cell-based measurement systems, such as pulsed-field gel electrophoresis (PFGE) or the detection of molecular responses to DNA damage via radiation induced foci (RIF).

One disadvantage was it is hard to compare our in vitro DNA analysis and cells directly because the DNA in cells has chromatin structure, which protects the DNA from radiation and is not in naked form [27]. Another disadvantage was that in order to appropriately determine the effects of each radiation source on the DNA in the presence of Cu^2+^ or Co^2+^, we had to utilize nearly three times the amount of single-strand DNA than double strand DNA in each of our 10 µL reaction solutions, 83 ng and 30 ng, respectively, as our agarose gel electrophoresis system was not sensitive enough to detect ssDNA bands at the lower concentration following low irradiation dosages particularly when in solution with Cu^2+^. Thus, the relative amount of SSBs may be underestimated due to the sensitivity of our assay if compared directly with the relative amount of DSBs at the same irradiation dosage. Lastly, electrophoresis-based detection DNA damage systems require much higher ionizing radiation doses than the usual radiobiologically relevant doses [28].

On the other hand, the main advantage of our system was the ability to isolate the DNA in solution. This allowed us to address how each metal ion influenced the amount of DNA damage via our proposed mechanism of Fenton-type reaction as well as eliminate other possible variables that arise from cell-based experiments, such as the radiation protection of the chromatin structures of cellular DNA. In addition, the cells’ abilities to regulate the intercellular amounts of each tested metal and the antioxidant capabilities within the cellular environments could also interfere with the Fenton-type reaction mechanism investigated in this study. Moreover, cell-based measurement systems typically utilize DNA repair proteins as surrogate markers of DNA damage, making time a confounding factor due to the delay between radiation and DSB detection/repair [28]. This would become problematic to our study as “simple” DNA damage may be repaired quickly and go undetected.

While it is difficult to directly compare our results to those within a cellular system, the utilization of metals as radiosensitizers is not a new concept. However, the mechanisms behind their radiosensitization effect in cells are not fully understood, particularly for gold nanoparticles (NPs) that have been of great interest clinically. Sabella et al. demonstrated that the toxicity mechanism for different metal-containing NPs is associated with the release of the corresponding toxic ions and that the protective cellular machinery designed to degrade the foreign objects is responsible for their toxicity [29]. In addition, Penninckx et al. suggested that gold nanoparticles play a radiosensitizer role by weakening detoxification systems [30]. Other studies also provide evidence that gold nanoparticles can enhance the radiation effects through an increase in reactive oxygen species production (such as hydroxyl radicals) [31]. Taken together, these prior studies support our mechanistic hypothesis that the metal ions, Cu^2+^ and Co^2+^, increase the DNA damage via Fenton-type reaction (Figure 4).

Using naked DNA in the presence of Cu^2+^ or Co^2+^, an increase in the amounts of DSBs and SSBs were observed for both low- and high-LET radiation (Figure 1). Interestingly, as the LET increased the fold reduction in MER values, it decreased for both metals in dsDNA, as well as with cobalt in ssDNA, but increased for copper in ssDNA (Figure 3). A possible explanation for these observed results may be due to copper’s ability over cobalt’s to more efficiently interact with the hydrogen peroxide produced from the high-LET irradiations via Fenton-type reaction and resulting in the production of more hydroxyl radical formation, thus increasing the amounts of more randomly distributed SSBs (Figure 4). Moreover, metal ions are essential for the formations of strand breaks with hydrogen peroxide through Fenton-type reaction to produce hydroxyl radicals (Figure 2b,d) [10], as well as site-specific mechanisms in the formation of DSBs mediated with Cu^2+^ Fenton reactions [7] (Figure 4). Moreover, our results agreed with prior research that the Cu^2+^ Fenton system generated a higher total yield of DNA lesions than with the Co^2+^ Fenton system [19].

Importantly, even though copper has an increased ability to induce hydroxyl radical ions through Fenton-type reaction, mammalian cells are not efficiently killed by SSBs caused by hydroxyl radicals from hydrogen peroxide. This is because hydroxyl radicals induced singly damaged sites are efficiently and accurately repaired by cellular repair mechanisms in contrast to hydroxyl radicals produced via the radiolysis of water causing LMDS, which are much more difficult for the cell to repair [9]. This is most likely due to DSBs being the major actor responsible for radiation-induced cell death, which occurs from localized SSBs [32]. Although, as our hydrogen peroxide results were consistent with our ionizing radiation results, in which copper in solution with DNA not only increased SSBs but DSBs as well, this may suggest that the hydrogen peroxide produced from the ionizing radiation, importantly from our high-LET radiation sources, was in close enough proximity to the DNA that the Fenton-type reaction production of hydroxyl radical ions can induce these LMDS. Future experiments examining direct plasmid analysis to confirm how copper influences high-LET radiation-induced DNA damage and their complexity would be beneficial to further support this reasoning [19].

Finally, we observed how copper and cobalt influence DNA breaks with bleomycin, a radiomimetic drug that binds with a metal and uses molecular oxygen to induce DSBs [33]. Prior studies have suggested that Cu^2+^–bleomycin complex is inactive in the degradation of DNA [34,35] while others have demonstrated that the Cu^2+^–bleomycin complex does indeed produce DNA strand scission [36,37]. Our results agree with the latter, in which copper ions in solution with bleomycin and DNA were observed to induce DSBs and SSBs more efficiently than in the solution with the control (Figure 2a,c). On the other hand, we observed a concentration dependent switch from sensitization to protection for Co^2+^ with bleomycin in both dsDNA and ssDNA. A possible reasoning behind this may be from the fact that it has been established that Co^2+^-bleomycin binds efficiently only at certain sites of DNA and that cleavage does not occur at all bound sites [38]. Furthermore, our observed results may also be explained as prior studies have demonstrated that as the ratio of DNA to Co^2+^–bleomycin complex increases, the Co^2+^–bleomycin complex becomes resistant to oxidation [36,37,38,39]. In addition, another study demonstrated that bleomycin forms an exchange–inert complex with cobalt that has a much higher affinity for DNA than most other metal bleomycin’s as well as little cytotoxic activity, which further supports our results [39]. Taken together, the MER values of bleomycin were the smallest among tested agents with metals. This suggested that bleomycin-induced DNA damage is not strongly associated with hydroxyl radicals.

## 5. Conclusions

The present work demonstrates that copper and cobalt may be useful tools to enhance the indirect action of DNA damage for high-LET irradiations. Both metals were observed to be capable of increasing DNA strand breaks following irradiation. However, copper was observed to be more efficient than cobalt at inducing these strand breaks. We propose the mechanism behind this observation is due to their interaction with hydrogen peroxide produced from the radiolysis of water via Fenton-type reaction, thus resulting in increased amounts of highly reactive hydroxyl radical ions. This was supported as we also observed this increase in DNA strand breaks with the metals in solution with hydrogen peroxide and DNA. These results suggest a possible mechanism of enhancement for the indirect action of DNA damage produced via high-LET radiation.

## Figures and Tables

**Figure 1 toxics-11-00773-f001:**
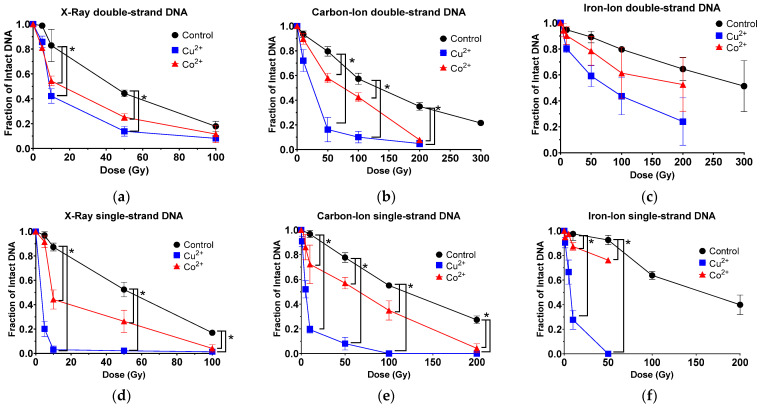
Cu^2+^ and Co^2+^ effect on DNA DSBs and SSBs at increasing radiation dosage (Gy) with radiation sources of increasing LET values. (**a**) Low-LET X-ray for DSBs; (**b**) high-LET carbon-ion for DSBs; (**c**) high-LET iron-ion for DSBs; (**d**) low-LET X-ray for SSBs; (**e**) high-LET carbon-ion for SSBs and (**f**) high-LET iron-ion for SSBs. Error bars indicate standard error of the means from at least three independent experiments. * indicates statistical differences (*p* < 0.05).

**Figure 2 toxics-11-00773-f002:**
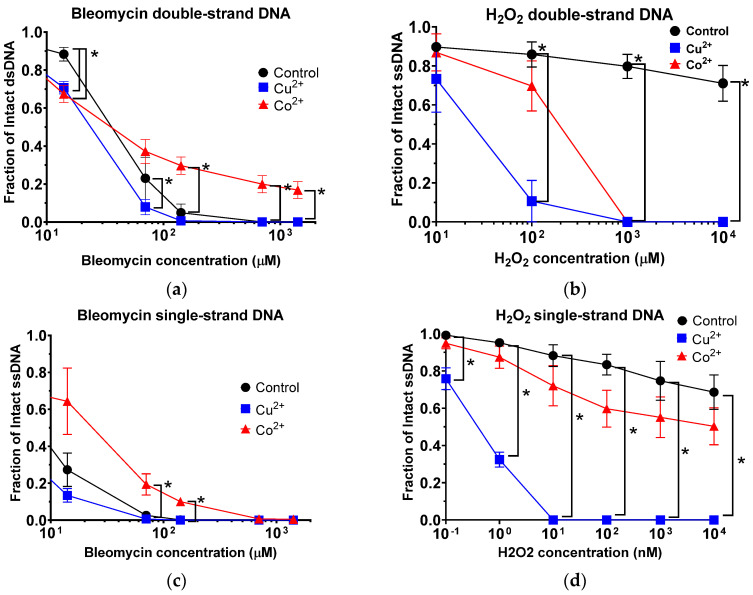
Cu^2+^ and Co^2+^ effect on DNA DSBs and SSBs at increasing drug concentrations of either Bleomycin or hydrogen peroxide. (**a**) Bleomycin for DSBs; (**b**) hydrogen peroxide for DSBs; (**c**) Bleomycin for SSBs and (**d**) hydrogen peroxide for SSBs. Error bars indicate standard error of the means from at least three independent experiments. * indicates statistical differences (*p* < 0.05).

**Figure 3 toxics-11-00773-f003:**
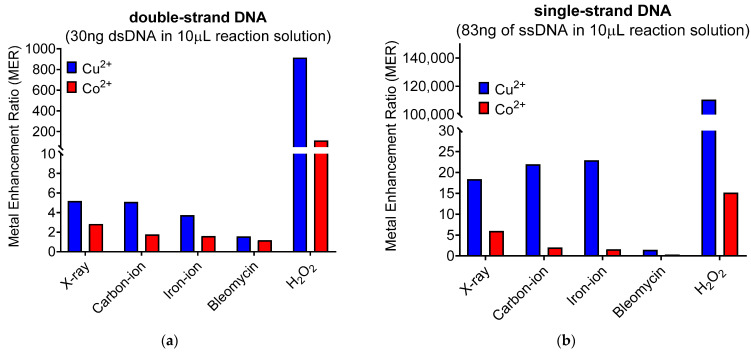
Metal enhancement ratio (MER) for DNA break formation comparison between ionizing radiation and chemical treatment with metals in solution with DNA (**a**) dsDNA; (**b**) ssDNA.

**Figure 4 toxics-11-00773-f004:**
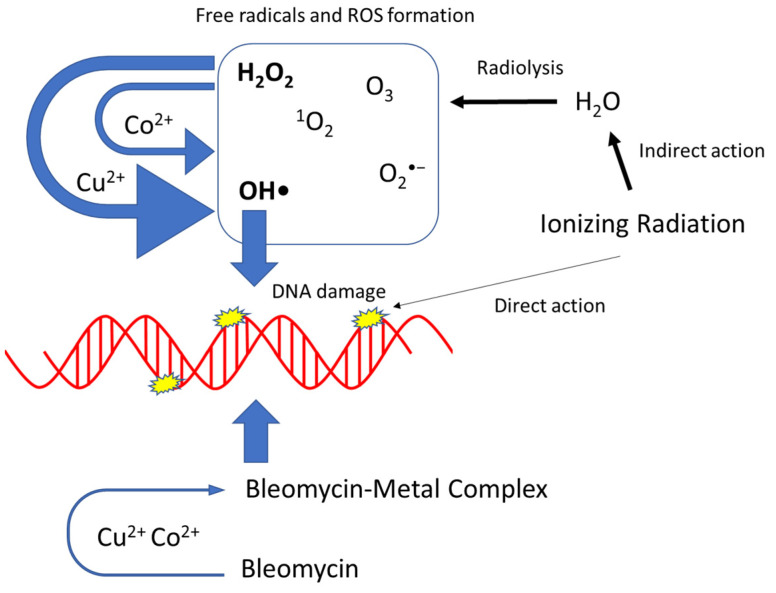
Proposed mechanisms of metal enhancement of ionizing radiation, bleomycin and H_2_O_2_. Thickness of arrows associated with the degree of effects.

**Table 1 toxics-11-00773-t001:** D_50_ and IC_50_ values, radiation doses (Gy) or chemical concentration required for 50% intact DNA with or without metal ions.

	Control	Cu^2+^	Co^2+^
DSB	Initial dsDNA amount: 30 ng Lambda phage dsDNA in 10 μL of reaction solution		
X-ray	44 Gy	8.5 Gy	15.5 Gy
Carbon-ion	142.9 Gy	28.1 Gy	80.9 Gy
Iron-ions	299.1 Gy	80 Gy	187 Gy
Bleomycin	46,625 nM	29,882.4 nM	39,560.6 nM
H_2_O_2_	20,073 nM	22 nM	177 nM
SSB	Initial ssDNA amount: 83 ng M13 bacteriophage ssDNA in 10 μL of reaction solution		
X-ray	57.5 Gy	3.1 Gy	9.6 Gy
Carbon-ion	129.8 Gy	2.9 Gy	65 Gy
Iron-ions	160 Gy	7.0 Gy	102.0 Gy
Bleomycin	8689.2 nM	4591.9 nM	26,138.2 nM
H_2_O_2_	17,237 nM	0.156 nM	1138 nM

## Data Availability

Data are contained within the article.

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
