# Peer review of "Metal Ions Modify In Vitro DNA Damage Yields with High-LET Radiation"

_toxics, 2023, doi:10.3390/toxics11090773_

Round 1

Reviewer 1 Report

In their paper entitled "Metal Ions Modify in-vitro DNA Damage Yields with High-LET Radiation", the authors investigate the impact of metal ions on DSB generation and the associated influence of particle's LET. The article is well written and presents interesting data. My only comments concern the discussion. 

1. The authors briefly discuss a limitation of the in vitro model used, which is the structure of chormatin. I would like to see this discussion extended more generally to the advantages and disadvantages of using their model versus cell-based measurements, which are the gold standard. I'm thinking in particular to the usual drawback of cell-based measurements, which is the phenomenon of DSB repair concomitant with the measurements, which  necessitate the use of mathematical models. S. Costes' group recently published guidelines for this type of experiment (doi.org/10.1093/narcan/zcab046). 

2. The discussion around figure 4 should be put into perspective within a more global cell biology framework. The explanation of increase DSB yield given is an increase in the generation of radical species produced by Cu2+ Fenton-type reactions. However, one might question the impact of such reactions in cells that usually possess antioxidant systems capable of counteract this increase in reactive oxygen species. Interestingly, metal ions are known to be inhibitors of these antioxidant systems, which would reinforce the paper's mechanistic hypothesis (see literature on this topic : doi.org/10.3390/cancers12082021; doi.org/10.1039/C4NR01234H, doi.org/10.2217/nnm-2018-0171). 

It would therefore be interesting to broaden the discussion by incorporating these elements. 

Reviewer 2 Report

Buglewicz et. al. shows how metal ions (such as Cu2+ and Co2+) are capable of increasing DNA strand breaks following different forms of irradiation or chemical treatments such as H2O2 or bleomycin. While it has been known that irradiation generated oxidative radicals and H2O2 together with cellular metal ions through Fenton-type reactions could enhance their effect on DNA, this was nicely confirmed by the authors in vitro. The Co2+ being protective against bleomycin delivered DNA strand break was interesting. Though the study in its current form is quite limited, I can accept that further work would be out of scope for the authors (such as repeating these experiments on in vitro packed histone-DNA octamers or including other radiomimetic drugs such as Neocarzinostatin (NCS), or if the authors could have also quantified the ssDNA nicks generated on the dsDNA substrate). I have no technical issues with the simple experiment that was performed by the authors.

Minor points:

-       - Please include the manufacturer of bleomycin and H2O2.

-    - Please include the exact length of DNA used in the study. The dsDNA on NEBs website contains the lengths, but not for the ssDNA.

-     - Use less cropped gel images in the supplementary materials as the pattern of the smearing of the DNA is also informative.

-        - Was there any reason that the authors never tried including Fe2+?

-    - Did the authors consider that Co2+ might show protection against bleomycin action because it outcompetes the iron it normally chelates?

Moderate editing of English language is required. Especially the discussion and conclusion parts need corrections.

Reviewer 3 Report

In the manuscript „Metal Ions Modify in-vitro DNA Damage Yields with High-LET Radiation“ the authors Buglewicz et al. present the effect of Cu2+ and Co2+ metal ions on the DNA damage yield caused by different DNA damaging agents.

The paper is not very extensive but presents interesting data on modifying the extent of DNA damage by the two cations. It would be interesting to check the relevance of these results in cellular environment.

However, the manuscript should be improved/brushed up by detailed English editing, as there are several misspellings and the language style should be improved in some parts. Below are some of the remarks that should be addressed:

In line 16 please add“-„ in Fenton type

In line 19 replace „was“ with „were“

In lines 21-24 the sentence should be improved, among other to change the end into „after treatment with bleomycin“

In line 26 add „-„ in LET dependent

In line 40 please remove „by“

In line 46 replace „radial“ with „radical“

In line 94-95 please replace „destained“ with „ destaining“

In the Section 2.3. in line 95-96 the authors should name the machine that was used for gel imaging and also the software that was used for intensity measurements.

In line 97 „or treated without chemicals“ should be replaced by „nor treated with chemicals“

In line 105 the sentence „All experiments…“ should be rewritten

The titles of the sections 3.1.-3.5. should be more descriptive of the results shown in particular section

In line 118 replace „compare“ with „compared“

In line 148 „damages“ should not be used in plural

In line 183-184 the sentence „On the other hand,…“ should be rewritten

In line 217 change „stand“ into „strand“

In line 267 the references 36-39 are cited, however the Reference list at the end of the manuscript contains only 33 references, and also, before the references 36-39 in the text the last reference number in line 264 is 33

The quality of the English language is mostly satisfactory, however the language style should be improved in certain parts and there are several misspellings.
